# Redox cycling of sulfur via microbes in soil boosts the bioavailability of nutrients to *Brassica napus*

Rabia Aslam[1], Ghulam Jilani[1]*, Tajwar Alam[1,2]*, Zia–Ul -Haq[3], Ambreen Bhatti[4], Riaz Ullah[5], Iram Naz[6], Muhammad Ikram[1], Nida Fatima[1], Essam A. Ali[7], Muhammad Zahoor[8], Shah Zaman[9]

**1** Institute of Soil and Environmental Sciences, PMAS Arid Agriculture University, Rawalpindi, Pakistan, **2** Department of Soil and Cop Sciences, Texas A&M AgriLife Research Center, Beaumont Texas A&M University, Beaumont, Texas, United States of America, **3** Faculty of Agriculture Engineering, PMAS-Arid Agriculture University, Rawalpindi, Pakistan, **4** Institute of Hydroponic Agriculture, PMAS Arid Agriculture University, Rawalpindi, Pakistan, **5** Department of Pharmacognosy, College of Pharmacy, King Saud University, Riyadh, Saudi Arabia, **6** Department of Agriculture Extension, Government of Punjab, Rawalpindi, Pakistan, **7** Department of Pharmaceutical Chemistry, College of Pharmacy King Saud University, Riyadh, Saudi Arabia, **8** Department of Biochemistry, University of Malakand chakdara, Malakand, Pakistan, **9** Department of Botany, University of Malakand, Malakand, Pakistan

* jilani@uaar.edu.pk (GJ); tajwaralam@uaar.edu.pk (TA)

## Abstract

Deficiency of sulfur (S) has been recognized for both dry and wetland plants as a significant growth limiting factor. S-oxidizers enhance the oxidation rate of S and increase sulfate ($SO_4^{-2}$) production by making them available in critical growth stages, resulting in higher plant growth and yield. A two-phase study was undertaken to assess the effectiveness of microbial redox reactions on soil nutrient bioavailability and crop growth. In the first phase isolation of S bacteria was carried out while in the second phase, a pot experiment was conducted and these bacteria were evaluated as a nutrient source along with different ratios of NPK fertilizer by growing canola (*Brassica napus*) as a test crop. Treatment combinations were, viz., Control (no inoculum), ½ NPK fertilizer (50, 30, and 25 kg ha$^{-1}$), Full NPK fertilizer (100, 60, and 50 kg ha$^{-1}$), ½ NPK+SOB, ½ NPK+SRB and ½ NPK+SOB+SRB. Sulfur oxidizing bacteria (SOB) strains were isolated from samples and then screened based on pH reduction (in thiosulphate broth). Sulfur-reducing bacteria (SRB) are characterized by their anaerobic energy metabolism based on the dissimilatory reduction of $SO_4^{-2}$ to hydrogen sulfide ($H_2S$) or S ° to be useful for SOB. Among different bacterial consortiums, the best results for nutrient bioavailability (macro- and micronutrient) in soil and plant the treatment ½ NPK+SOB+SRB compared to full NPK increased soil N, P, K, and $SO_4$ by 15.9%, 38%, 2.0%, and 72%, respectively. In plants, it boosted N, K, and $SO_4$ by 7.7%, 31%, and 239%, respectively. The greatest pH reduction (4%) was observed with ½ NPK+SOB relative to full NPK. This study demonstrates that microbial inoculation along with ½ NPK increases the bioavailability of macro- and micronutrients for crops.

**Data availability statement:** All relevant data are within the manuscript.

**Funding:** The authors extend their appreciation to the researchers supporting Project number (RSP2025R45) King Saud University, Riyadh, Saudi Arabia for financial support. This funding was received by EAA

**Competing interests:** The authors declare that they have no known competing financial interests or personal relationships that could have appeared to influence the work reported in this paper

## 1. Introduction

Sulfur is among the essential nutrients required for the proper growth of plants, animals, humans, and microorganisms [1]. Being a key component of amino acids S is vital for the formation of amino acids like methionine and cysteine, which are fundamental for plant growth and the synthesis of proteins and vitamins [2]. Sulfur plays a key role in forming vitamins, proteins, and oils improving plants compounds. A deficiency in S impairs photosynthetic activity, disrupts nitrogen (N) metabolism, reduces oil content, and hampers overall plant growth, with pronounced effects on both shoot and root development [3]. Its deficiency causes a decrease in S-containing amino acids and protein synthesis [4]. Additionally, its deficiency causes the yellowing of younger leaves known as chlorosis followed by necrosis in later developmental stages. Lower S availability also affects N fixation because both N and S are central parts of the protein [5]. Oilseed crops, viz., soybean, groundnut, rapeseed, and sunflower are required more S, followed by cereals and pulses [6]. Therefore, to increase S availability, S-oxidizers are used to improve the natural oxidation rate and enhance the production of $SO_4^{-2}$, making it available to crop plants at critical stages [7]. The application of mineral fertilizer having S has also been reported to improve nutrient availability, by improving soil physicochemical properties [8].

Application of S increased the synthesis of amino acids and also enhanced amounts of $N_2$ fixed in leguminous plants and soil [9]. The availability of S in the soil is affected by physiochemical factors and pedogenic processes. Most soils across the globe are S-deficient [10]. Sulfur deficiency is more common compared to other nutrients in soils of Northern Europe with oilseed cropping [11]. Additionally, S deficiency could also be attributed to decreased S storage via deposition from the atmosphere in the last two to three decades. Furthermore, S-containing fertilizers like single superphosphate, farmyard manure, and compost have been substituted by chemical fertilizers having no or little S [12].

Beneficial microorganisms from the rhizosphere and phyllo-sphere were isolated and evaluated as plant growth promotors to reduce agrochemicals application in soil [2]. Mechanisms directly involved in plant growth promotion relate to higher nutrient acquisition through fixation, solubilization in soil (N fixation, P, K, and S solubilization), hormone production (auxins, cytokinin, aminolevulinic acid, gibberellins, and abscisic acid) and iron sequestration through bacterial siderophores, and ACC deaminase synthesis to reduce formation of ethylene [13]. Whereas, indirect growth stimulation mechanisms include, the reduction of stresses viz., salinity, drought, heavy metals, and temperature [14]. Microbes possessing plant growth-promoting potential have been commercialized as biofertilizers, viz., N-fixing, P-solubilizing, K-mobilizers, PGPR, and mycorrhizal fungi [15], and inoculation of such microbes alter microbial diversity and change rooting patterns that help in nutrients management [16]. Microbes involved in S cycling could be used as bio-fertilizers, having low-input and environment-friendly technology for sustainable agriculture ecosystems [17].

Plants absorb S in the form of inorganic $SO_4^{2-}$ and some microbes can oxidize S into $SO_4^{-2}$ form known as SOB [18]. Bacteria belonging to genus Thiobacillus and Acidithiobacillus, are involved in S-oxidation (Shinde et al., 2022). Several studies have been undertaken using SOB as a microbial inoculant and results showed around 47–69% of onion yield increase compared to control [1]. Moreover, SOB inoculation and N-fixing bacteria combinedly improved plant yield and N uptake (220%, and 630%, respectively) compared to non-inoculated plants. Similarly, bio-fertilization has N-fixing strains (*Azotobacter* and *Azospirillum*), in addition to the inorganic N-enhanced oil content and grain yield of canola [19]. SOB enhanced S-oxidation resulting in higher availability of $SO_4^{-2}$ to mustard crop [20].

Sulfur bacteria integrate a diverse group of organisms having the capability to share oxidized, reduced, or partially oxidized inorganic S compounds. Genus *Thiobacillus* is the most

important organism in different groups of bacteria (SOB) which are responsible for S oxidization. Application of *Thiobacillus* bacteria enhances S and P availability in soil. Canola is the most essential oil seed crop across the globe [21]. P-solubilizing and SOB enhance canola efficiency in calcareous soils by improving the absorption of plant nutrients. Bacterial inoculant operations can be improved by the addition of organic matter (OM). Therefore, the correct combination of chemical and biological sources can considerably boost canola production and development by improving nutrient absorption [22]. Keeping in view the S importance as a key macronutrient, further studies are needed on soil microorganisms involved in their biogeocycle. The current study's novelty lies in a comprehensive evaluation of the effects of SOB and SRB in combination with NPK fertilizer's recommended dose synergistically. This approach augments our understanding of how interactions of microbes enhance nutrient bioavailability in soil which offers a novel approach to optimize the management of nutrients in crop cultivation.

Current study objective was to evaluate the impact of the recommended dose of NPK with or without SOB and the combined effects of SOB and SRB along with the recommended dose of NPK. It was inferred that their integrated application synergistically affects macro- and micronutrient bioavailability and uptake by canola in soil.

## 2. Materials and methods

### 2.1. Soil characterization

Ten composite samples of soil (0–30 cm) were collected at the start of the experiment to analyze soil pH, EC [23], OM [24], and texture [25]. The texture of experimental soil was recorded as silty clay loam, having pH = 7.53, EC = 0.252 dSm$^{-1}$, N (0.54 g kg$^{-1}$), P (6.91 mg kg$^{-1}$), K (131 mg kg$^{-1}$), S (6.94 mg kg$^{-1}$) zinc (0.31 mg kg$^{-1}$), manganese (4.01 mg kg$^{-1}$), and iron (4.2 mg kg$^{-1}$).

### 2.2. Isolation of SOB and SRB

SOB isolation was carried out with a thiosulphate broth medium. Its composition was, viz., $Na_2S_2O_3$, $NaHCO_3$, 0.2 g; 5.0 g; 0.1 g; $NH_4Cl$, $K_2HPO_4$, 0.1 g dissolved in distilled (DI) water (DI) (1.0 L). Medium pH was adjusted to 8.0 and Bromocresol purple was used as indicator. The medium was autoclaved for sterilization and subsequently poured into pre-sterilized tubes and upon condensation, the streaking was done to purify the isolated strains. The tubes were incubated for 4–5 days at 30 °C [26].

Enrichment and isolation of SRB were done by using a medium containing DI water per liter: ammonium sulfate 5.3 g, sodium-acetate 2.0 g, $KH_2PO_4$ 0.5 g, magnesium sulfate.7$H_2O$ 0.2 g, sodium chloride 1.0 g, calcium chloride. 2$H_2O$ 0.1 g. Solution: 1 10.0 mL and Solution: 2 1.0 mL. Solution one having per liter of DI water: Nitrilotriacetic acid 12.8 g, $FeCl_2$.4$H_2O$ 300.0 mg, copper chloride 20.0 mg, $MnCl_2$.4$H_2O$ 100.0 mg, $COCl_2$.6$H_2O$ 170 mg, zinc chloride 100 mg, $H_3BO_3$ 10.0 mg, $Na_2MoO_4$.2$H_2O$ 10 mg. Solution two has DI water 100 mL Resazurin 0.2 g. Followed by autoclave and cooling, the medium was provided with anaerobic sterile stock solutions from components: 50 mL of 8% $Na_2CO_3$ water. 5.5 mL of 25% hydrochloric acid. About 1.0 mL of 8.7% $Na_2S_2O_4$ in water and, pH was adjusted to 7.2 by adding HCl [27].

### 2.3. Purification of SOB and SRB

Isolate purification was carried out by moving the isolates to the medium of the new broth. The streaking of isolates was carried out on thiosulfate ($S_2O_3^{2-}$) agar plates to obtain individual colonies. For characterization and further testing, these pure isolates have been maintained [28].

## 2.4. Characterization of SOB and SRB

Isolated strain characterization was done through colony morphology, elevation pattern, colony margins, colony colour, colony form, and opacity. Additionally, biochemical and morphological characteristics were studied to characterize isolated bacterial strains by following [29].

## 2.5. Gram staining

Bacterial strains were further subjected to gram staining as explained by [30]. Wire loop was first heated on a spirit lamp then it was full of individual isolated bacterial strains that were spread on a glass slide followed by air-drying and stained by using crystal violet for two minutes followed by slight washing with deionized water. Later on, the smear was flooded with iodine solution and de-colorized by using 75% alcohol. After de-colorization, the smear was stained with safranin. The smear was dried by passing a glass slide 2 to 3 times from spirit lamb and the slide was placed under a light microscope for observation of the staining reaction of individual isolates.

## 2.6. Treatment plan and experimental design

A greenhouse experiment was undertaken at PMAS-Arid Agriculture University Rawalpindi, to assess the potential impact of inoculation of SOB and SRB, and synthetic fertilizer on canola production, soil nutrients (macro- and micronutrient) bioavailability, and plant uptake. Soil was collected from the university research area and pots were filled (8 kg each) with air-dried and sieved soil (2 mm). The treatment combination was control, NPK half dose (½ NPK) (50–30–25 mg kg$^{-1}$), full dose of NPK (100–60–50 mg kg$^{-1}$), ½ NPK + SOB, ½ NPK + SRB, and ½ NPK + SOB + SRB. A completely randomized design (CRD) was implemented with three replications. Treatments encompassing bacteria and synthetic fertilizer were added to the soil before the sowing of the canola crop. Synthetic fertilizer (NPK) was applied as a basic dose before sowing as DAP, urea, and $K_2SO_4$, in all pots. Sterilized DI water was used to dilute bacterial inoculums at the rate of 1% v/v. Ten seeds of *Brassica napus* per pot were sown in November 2021. After seedlings establishment, thinning of plants was done (five plants$^{-1}$). Soil moisture (70–80%) was maintained and weeds were manually removed wherever required.

## 2.7. Crop harvesting

Harvesting of canola was done after 145 days of sowing. Different attributes, viz., shoot length, fresh shoot - and dry weight, and root fresh- and dry weight were recorded. The shoot and root dry weight of each plant was determined by separating roots from shoots with DI water and stored in an oven at $65 \pm 3$ °C for 3 days.

## 2.8. Analysis of soil

To evaluate the impact of bacterial inoculates and synthetic fertilizer on S and other elements, post-harvest soil was collected and analyzed for total-N [31], AB-DTPA available-P, and extractable-K [32]. AB-DTPA extractable-Mn, Zn, and Fe were recorded by following [33]. Soil texture was measured through the hydrometer method [25]. Soil pH and EC were measured through soil pH and EC meter [23]. Soil organic matter was analyzed by following [34].

## 2.9. Plant analysis

Canola plant leaves were harvested at the maturity stage, and dried in the oven at $65 \pm 3$ °C for 3 days, and dry weight was noted. Dried shoots/leaves were ground to powder and dry ashing

was done up to 4 h at 550 °C through a muffle furnace. Digestate was used to analyze Mn, Fe, and Zn at different wavelengths, 279.5 nm, 248.7 nm, and 213.7 nm respectively, and analyzed by using an atomic absorption spectrophotometer. Potassium contents were analyzed through a flame photometer. Phosphorous contents were measured by following [35]. Kjeldahl method was used to measure total-N [31].

### 2.10. Statistical analysis

A completely randomized design (CRD) was implemented with six treatments and replicated thrice. All data was analyzed by using Statistix 8.1. ANOVA and multiple comparison analyses were performed using the Tukeys test at $P < 0.05$. Means were compared by using the least significant difference test ($LSD_{0.05}$) for treatments' statistical significance. Graph was drawn by using MS Excel, 2010.

## 3. Results

### 3.1. SOB and SRB isolate attributes

Isolated S-oxidizing bacteria was *Thiobacillus thiooxidans*, a gram-negative chemo-lithotroph bacteria. They utilize $S_2O_3^{2-}$ and sulfide as energy sources to produce sulphuric acid. These are aerobic sulfur bacteria and they derive energy from the oxidation of sulfide or elemental sulfur ($S^0$) to $SO_4^{-2}$. The isolated S-reducing bacteria was Desulfvibrio vulgaris, Gram-negative, non-spore-forming, anaerobic, curved rod-shaped PGPR.

### 3.2. Growth attributes of canola

Plant attributes like root, shoot, and plant biomass of canola differed significantly with bacterial inoculation in soil. Results indicated that inoculation of SOB and SRB enhanced canola biomass compared with sole NPK application. The highest shoot length of 100 cm and the highest root length of 26.8 cm were noted in T6 (½ NPK+SOB+SRB) (Table 1). The highest fresh root weight (g plant⁻¹) of 2.58 was recorded in T5 (½ NPK + SRB), while the highest fresh shoot weight (g plant⁻¹) of 28.6 was noted in T6 (½ NPK + SOB + SRB). The highest dry root weight (g plant⁻¹) of 2.03 was noted in T6 (½ NPK + SOB + SRB), while the highest dry shoot weight (g plant⁻¹) of 15.9 was obtained in T6 (½ NPK + SOB + SRB) (Table 1).

### 3.3. Nutrient contents in canola

The concentration of nutrients in plant tissue at maturity depicted a significant response to applied treatments. The highest total-N (1.53%), K (2.80%) concentration was recorded in T6

**Table 1. Effect of microbial inoculants and fertilizer application on crop growth parameters.**

| Treatment | Shoot length (cm) | Root length (cm) | Fresh shoot weight (g) | Dry shoot weight (g) | Fresh root weight (g) | Root dry weight (g) |
|---|---|---|---|---|---|---|
| Control | 75.9 ± 2.96 e | 17.3 ± 1.86 d | 11.4 ± 1.32 d | 8.00 ± 0.10 e | 1.31 ± 0.03 e | 0.90 ± 0.10 e |
| 1/2 NPK | 84.1 ± 2.33 d | 19.1 ± 1.18 cd | 19.9 ± 1.08 c | 10.2 ± 0.94 d | 1.93 ± 0.07 d | 1.39 ± 0.16 d |
| full NPK | 90.5 ± 0.80 c | 20.6 ± 1.64 c | 23.8 ± 3.43 b | 13.0 ± 0.94 c | 2.26 ± 0.05 c | 1.61 ± 0.17 c |
| 1/2 NPK+SOB | 94.9 ± 1.82 b | 23.6 ± 1.52 b | 25.6 ± 1.96 b | 14.4 ± 0.93 b | 2.41 ± 0.29 b | 1.83 ± 0.16 b |
| 1/2 NPK+SRB | 90.9 ± 1.96 c | 22.7 ± 1.01 b | 21.8 ± 1.39 c | 12.2 ± 1.43 c | 2.32 ± 0.05 bc | 1.68 ± 0.04 c |
| 1/2 NPK+SOB+SRB | 100 ± 0.88 a | 26.8 ± 1.31 a | 28.6 ± 1.49 a | 15.9 ± 0.42 a | 2.58 ± 0.53 a | 2.03 ± 0.25 a |

Each value is a mean of three replicates ± SD (n = 3). Different letters indicate significant differences between treatments within columns by LSD test (P ≤ 0.05). ½ NPK, synthetic fertilizer @ 50–30–25 kg ha⁻¹; NPK, synthetic fertilizer @ 110–60–50 kg ha⁻¹; SOB, S-oxidizing bacteria; SRB, S-reducing bacteria

(½ NPK+SOB+SRB) while the highest P (1.49%) concentration was recorded in T2 (full NPK) as reported in Fig 1. The highest total-S (0.21%) was recorded in T6 (½ NPK+SOB+SRB) followed by treatment 1/2 NPK+SOB (0.18%) while the lowest total-S (0.043%) was recorded for the control treatment (Fig 1). A blend of ½ NPK synthetic fertilizer with SOB and SRB (T6) improved Mn significantly (0.06 g kg⁻¹), Fe (0.023 g kg⁻¹), Zn (0.045 g kg⁻¹), and Cu (0.092 g kg⁻¹) contents in plants tissue as depicted in Table 2. Inoculation of SOB and SRB with half NPK, improved nutrient contents in canola crops significantly, which suggests these microorganisms' role in nutrient mobilization.

### 3.4. Post-harvest soil pH, EC, and OM contents

Post-harvested soil amended through bacteria significantly affected pH and OM. The decreasing pattern was shown in pH with various applied treatments, ranging from 7.5 in the control treatment to 7.1 in T4 (1/2 NPK + SOB) closely followed by T6 (7.2) ½ NPK+SOB+SRB (Table 3). However, OM and EC were increased slightly by different treatment applications. Soil OM content was recorded at 0.50% in control and 0.61% in treatment T6 (1/2 NPK+SOB+SRB). Soil EC was increased from 0.251 dS m⁻¹ in T1 (control) treatment to 0.492 dS m⁻¹ in T6 (1/2 NPK+SOB+SRB) (Table 3).

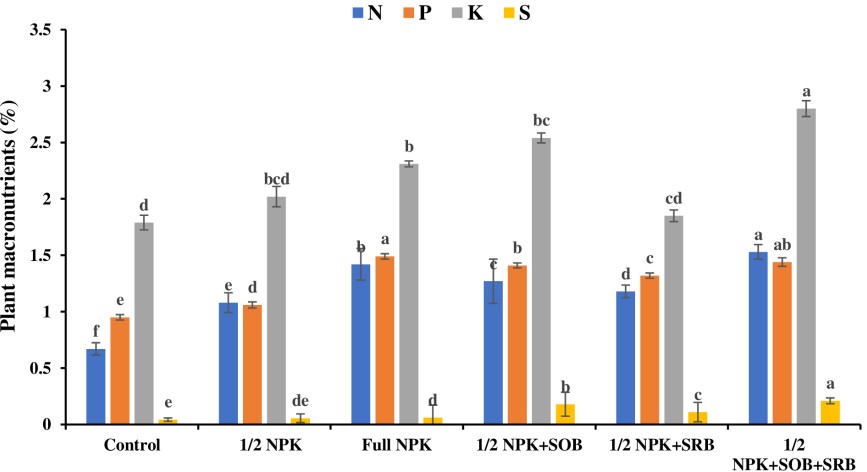

**Fig 1. Effect of microbial inoculants and fertilizer application on plant micronutrients.**

**Table 2. Effect of microbial inoculants and fertilizer application on plant micronutrients.**

| Treatment | Cu in Plant (mg kg⁻¹) | Mn in plant (mg kg⁻¹) | Fe in plant (mg kg⁻¹) | Zn in plant (mg kg⁻¹) |
|---|---|---|---|---|
| **Control** | 4.66 ± 0.05 f | 29.7 ± 4.3 e | 146 ± 7.22 f | 28.8 ± 0.53 e |
| **1/2 NPK** | 5.36 ± 0.15 e | 34.3 ± 3.5 d | 171 ± 14.7 e | 32.6 ± 2.55 d |
| **full NPK** | 6.40 ± 0.40 d | 39.0 ± 3.3 c | 190 ± 5.57 d | 36.1 ± 1.13 c |
| **1/2 NPK+SOB** | 8.86 ± 0.96 b | 54.2 ± 1.3 b | 261 ± 1.94 b | 43.3 ± 3.05 ab |
| **1/2 NPK+SRB** | 8.23 ± 0.20 c | 55.1 ± 3.8 b | 243 ± 20.2 c | 40.8 ± 1.09 b |
| **1/2 NPK+SOB+SRB** | 9.20 ± 0.26 a | 60.0 ± 3.3 a | 283 ± 0.61 a | 45.1 ± 2.51 a |

Each value is a mean of three replications ± SD. Different alphabets show significant differences between treatments within columns by LSD test (P ≤ 0.05). ½ NPK, synthetic fertilizer @ 50–30–25 kg ha⁻¹; NPK, synthetic fertilizer @ 110–60–50 kg ha⁻¹; SOB, S-oxidizing bacteria; SRB, S-reducing bacteria

## 3.5. Nutrient contents in post-harvest soil

Available-P concentration significantly improved over control (0.057 g kg$^{-1}$) with treatments applied up to the highest rate of 0.025 g kg$^{-1}$ at T6 (½NPK+SOB+SRB). Soil extractable-K and total-N also improved. Total-N varied between 0.56% to 1.53% showing that total-N was slightly short in post-harvest soil. Soil extractable-K varied from 1.31% to 1.56%. Soil S ranged from 6.73% to 13.3% (Table 4). Compared to control significant improvement in soil Cu, Mn, Fe, and Zn was recorded in treatment T6 (½NPK+SOB+SRB) with highest values of 0.32, 1.49, 2.54, and 1.33 mg kg$^{-1}$, respectively (Table 5).

**Table 3.** Effect of microbial inoculants and fertilizer application on soil chemical parameters.

| Treatment | Soil pH | EC (dSm$^{-1}$) | Organic matter (%) |
|---|---|---|---|
| **Control** | 7.5 ± 0.09 a | 0.251 ± 0.02 f | 0.50 ± 0.16 c |
| **1/2 NPK** | 7.5 ± 0.11 a | 0.283 ± 0.03 e | 0.55 ± 0.19 b |
| **full NPK** | 7.5 ± 0.16 a | 0.371 ± 0.02 d | 0.61 ± 0.09 a |
| **1/2 NPK+SOB** | 7.1 ± 0.12 c | 0.419 ± 0.02 b | 0.50 ± 0.16 c |
| **1/2 NPK+SRB** | 7.5 ± 0.04 a | 0.401 ± 0.04 c | 0.55 ± 0.19 b |
| **1/2 NPK+SOB+SRB** | 7.2 ± 0.04 b | 0.492 ± 0.08 a | 0.61 ± 0.19 a |

Each value is a mean of three replications ± SD. Different alphabets show significant differences between treatments within columns by LSD test (P ≤ 0.05). ½ NPK, synthetic fertilizer @ 50–30–25 kg ha$^{-1}$; NPK, synthetic fertilizer @ 110–60–50 kg ha$^{-1}$; SOB, S-oxidizing bacteria; SRB, S-reducing bacteria

**Table 4.** Effect of microbial inoculates and fertilizer application on soil macronutrients.

| Treatments | N (%) | P (mg kg$^{-1}$) | K (mg kg$^{-1}$) | SO$_4$ (mg kg$^{-1}$) |
|---|---|---|---|---|
| **Control** | 0.56 ± 0.06 d | 5.73 ± 2.00 f | 131 ± 1.84 c | 6.73 ± 0.18 e |
| **1/2 NPK** | 1.13 ± 0.12 c | 9.87 ± 0.31 e | 142 ± 1.12 b | 7.01 ± 0.25 e |
| **full NPK** | 1.32 ± 0.21 b | 17.7 ± 1.90 c | 151 ± 3.15 a | 7.72 ± 0.41 d |
| **1/2 NPK+SOB** | 1.49 ± 0.09 a | 21.0 ± 1.06 b | 156 ± 1.55 a | 12.7 ± 0.85 b |
| **1/2 NPK+SRB** | 1.12 ± 0.06 c | 14.7 ± 4.46 d | 149 ± 5.97 ab | 10.8 ± 1.01 c |
| **1/2 NPK+SOB+SRB** | 1.53 ± 0.07 a | 24.5 ± 3.00 a | 154 ± 3.21 a | 13.3 ± 0.76 a |

Each value is a mean of three replications ± SD. Different alphabets show significant differences between treatments within columns by LSD test (P ≤ 0.05). ½ NPK, synthetic fertilizer @ 50–30–25 kg ha$^{-1}$; NPK, synthetic fertilizer @ 110–60–50 kg ha$^{-1}$; SOB, S-oxidizing bacteria; SRB, S-reducing bacteria

**Table 5.** Effect of microbial inoculates and fertilizer application on soil micronutrients.

| Treatments | Cu (mg kg$^{-1}$) | Mn (mg kg$^{-1}$) | Fe (mg kg$^{-1}$) | Zn (mg kg$^{-1}$) |
|---|---|---|---|---|
| **Control** | 0.21 ± 0.01 c | 0.79 ± 0.12 e | 1.22 ± 0.68 d | 0.65 ± 0.04 d |
| **1/2 NPK** | 0.22 ± 0.06 c | 1.01 ± 0.09 d | 1.37 ± 0.12 d | 0.73 ± 0.04 d |
| **Full NPK** | 0.27 ± 0.04 b | 1.20 ± 0.22 c | 1.78 ± 0.20 c | 0.94 ± 0.01 c |
| **1/2 NPK+SOB** | 0.33 ± 0.04 a | 1.37 ± 0.28 b | 2.41 ± 0.77 a | 1.25 ± 0.05 a |
| **1/2 NPK+SRB** | 0.30 ± 0.01 ab | 1.36 ± 0.42 b | 1.99 ± 1.05 b | 1.17 ± 0.13 b |
| **1/2 NPK+SOB+SRB** | 0.32 ± 0.02 a | 1.49 ± 0.09 a | 2.54 ± 0.44 a | 1.33 ± 0.09 a |

Each value is a mean of three replications ± SD. Different alphabets show significant differences between treatments within columns by LSD test (P ≤ 0.05). ½ NPK, synthetic fertilizer @ 50–30–25 kg ha$^{-1}$; NPK, synthetic fertilizer @ 110–60–50 kg ha$^{-1}$; SOB, S-oxidizing bacteria; SRB, S-reducing bacteria

## 4. Discussion

The present study reinforced the above-stated hypothesis and revealed that the application of SOB and SRB along with chemical fertilizer highly influenced the physicochemical attributes of soil, enhanced canola growth, and increased bioavailability of nutrient contents in soil. Findings of the current study are in accordance with literature published previously [36], which reported that improvement in crop parameters is because of the release of bacterial metabolite and nutrient mineralization. The increase in crop shoot-root length and plant biomass was owing to the production of exopolysaccharides, siderophores, and phytohormones, and enzyme activation by *Leptothrix discophora* and *Bacillus polymyxa* [37]. Biofertilizer application enhances the growth of plants by improving the availability of nutrients in the rhizosphere through the production of antibiotics and by hindering pathogenic bacteria growth [38].

Inoculation of SOB and SRB with synthetic fertilizer significantly improved growth and yield attributes of canola compared to control [39]. Our findings also revealed that nutrient augmentation was much higher when the SOB and SRB inoculants were applied with half NPK recommended dose however, other studies [40], found limited results with similar microorganisms in acidic soils, and the difference in results may be attributed to differences in soil characteristics and environmental conditions. Application of Bacillus spp. improved micronutrients, viz., Mn, Zn, and Fe in plants [41]. Soil microbes contribute to solubility and henceforth improve soil micronutrients [42]. The impact of PGPR on plant production is well documented and has been attributed to the synthesis of phytohormones and a greater supply of nutrients [43]. Microbes use several methods to enhance the solubility of nutrients in soil like as altering plant metabolism and changing root exudates [44–45]. Microbial inoculation of *Bacillus mucilaginous* and *Bacillus megaterium* improved plant growth [46].

Oxidation and reduction of S by microbes are the most active processes in the S-cycle carried out by SOB and SRB and are considered vital phenomena in S biogeochemical cycling. Generally, on a nutritional basis, SRB and SOB are characterized as litho-autotrophs. Reduced compounds of S are oxidized by SOB like $H_2S$, $S^0$, sulfite ($SO_3^{-2}$), $S_2O_3^{2-}$, and $SnO_6^{2-}$ or $-SnO_6^{-}$ into $SO_4^{-2}$. However, $SO_4^{-2}$ serves as an SRB electron acceptor in anaerobic conditions and reduces $SO_4^{-2}$ and other S compounds ($S_2O_3^{2-}$, $SO_3^{-2}$, $S^0$) into $H_2S$. Moreover, in a natural ecosystem, SRB reduces $SO_4^{-2}$ through assimilatory and dissimilatory reactions. SRB utilizes various types of enzymes in dissimilatory reactions to reduce S substrate, while $SO_4^{-2}$ is assimilated in organic compounds via an assimilatory process through S substrate reduction. The soil used in the present study was slightly calcareous; both SOB (*Thiobacillus thiooxidans*) and SRB (*Desulfvibrio vulgaris*) inoculation significantly reduced soil pH. This might be because of organic acid production in the rhizosphere by soil microorganisms. Bacteria produce the few organic acids in soil called carboxylic acids [47], which lower soil pH in the rhizosphere and dissociate calcium phosphate bonds in calcareous soils. Furthermore, microbes modify redox potential and surrounding medium pH [48]. Our results are also in line with [49], who stated that a living organism's presence in soil produces pH variation and momentous redox potential within the soil. SOB or SRB has an important role as they significantly affect pH and redox potential [48]. Plant root's exudation of protons ($H^+$), carboxylates, and enzymes also affects soil pH [50]. SOB and SRB improve fertility of the soil by regulating pH and EC. During this study, microbial inoculate application slightly increased OM content of the soil. *Bacillus polymyxa* and *Thiobacillus thiooxidans* increase OM in soil via the release of numerous exopolysaccharides that break down large polymeric substances which in turn offer nutrients to plants [51].

Nutrient availability is controlled by pH and redox conditions [52]. Additionally, soil microfauna is also a key factor for nutrient dynamics in the soil [47]. We noted that soil

post-harvest nutrients (macro- and micro) responded greatly to SOB and SRB inoculation in combination with ½ NPK fertilizer. The highest uptake of nutrients by plants was recorded at pH 7.0, even with the lowest chemical fertilizer dose. Similar results have been reported by other researchers. According to [53], rhizosphere acidification and salinization are the most significant factors affecting the availability of nutrients in soil. Inoculation of seeds with *Leptothrix discophora* and *Bacillus polymyxa* bacteria enhanced solubilization as well as mineralization of macronutrients [54]. Several bacterial species play a vital role in increasing soil fertility through increasing OM, which enhances the availability of macronutrients in soil [1]. Moreover, by producing organic acids they contribute to nutrient mobilization and uptake by plants. The population of SOB and SRB is a key factor in the availability of soil S for plants [55]. Microorganisms mobilize S in aerated soils via SOx reduction that promotes $H^+$ root excretion, decreases the pH of saline soils, and improves the availability of micronutrients [1]. Availability of soil S is affected by the pH of the soil and improves greatly owing to decreases in pH. It could be due to S oxides becoming stronger oxidants when pH of the soil decreases, resulting in more easily reduced S ions [56].

However, excess application of S may acidify the soil, decreasing the availability of P by promoting their fixation which enhances competition between phosphate and $SO_4^{-2}$ ions. Further, it enhances K leaching, particularly in soils with lower CEC, and disrupts nutrient approval balance by favouring $SO_4^{-2}$ absorption. Eventually, degrading soil health reduces the efficiency of P and K, which necessitates supplementary fertilizer to correct this imbalance. Farmers can take advantage of the current study by exploiting the microbial redox cycling of S to improve the bioavailability of nutrients, increasing *Brassica napus* growth and yield while potentially decreasing fertilizer costs.

## 5. Conclusion

Findings of the current study revealed that the application of bacteria (S-oxidizing and S-reducing) with ½ NPK fertilizer rendered higher S availability as well as nutrients to canola by decreasing S immobilization in soil. Treatment ½ NPK+SOB+SRB improved soil N, P, K, and $SO_4$ by 15.9%, 38%, 2.0%, and 72%, respectively, and enhanced plant N, K, and $SO_4$ by 7.7%, 31%, and 239%, compared to full NPK. Additionally, ½ NPK+SOB showed the highest pH reduction (4%). Furthermore, significant improvement in the fertility status of soil was noted. In conclusion, these results suggest that combined application of synthetic fertilizers along with SOB and SRB inoculation as a soil amendment improves plant growth attributes, and nutrients in plants and soil. It is easily adaptable by farmers and eco-friendly methods to reduce crop nutrient rations, high-yield production, and sustain satisfactory profit. The study highlighted the benefits of SOB and SRB inoculation, suggesting future research on their effectiveness across soil types, climatic conditions, and interactions with fertilizers to optimize resource use in canola cultivation.

AcknowledgmentThe authors sincerely thank the Institute of Soil & Environmental Sciences, PMAS-Arid Agriculture University, Rawalpindi, for providing laboratory facilities and greenhouse space for analysis and experimentation.

## Author contributions

**Conceptualization:** Rabia Aslam, Ghulam Jilani, Tajwar Alam, Nida Fatima.

**Data curation:** Ghulam Jilani, Tajwar Alam.

**Formal analysis:** Rabia Aslam, Muhammad Ikram, Nida Fatima.

**Funding acquisition:** Essam A. Ali.

**Investigation:** Rabia Aslam, Tajwar Alam, Zia-Ul -Haq, Muhammad Ikram.

**Methodology:** Rabia Aslam, Tajwar Alam, Muhammad Ikram, Nida Fatima.

**Project administration:** Ghulam Jilani, Tajwar Alam.

**Resources:** Ghulam Jilani, Tajwar Alam, Zia-Ul -Haq.

**Software:** Rabia Aslam.

**Supervision:** Ghulam Jilani, Tajwar Alam.

**Validation:** Tajwar Alam.

**Visualization:** Tajwar Alam, Zia-Ul -Haq.

**Writing – original draft:** Rabia Aslam, Ghulam Jilani, Tajwar Alam, Nida Fatima.

**Writing – review & editing:** Zia-Ul -Haq, Ambreen Bhatti, Riaz Ullah, Iram Naz, Muhammad Ikram, Essam A. Ali, Muhammad Zahoor, Shah Zaman.

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
