## [Decision Letter · Decision Letter 0]

4 Dec 2024

PONE-D-24-23262Redox cycling of sulfur via microbes in soil boosts the bioavailability of nutrients to Brassica napusPLOS ONE

Dear Dr. Alam,

Thank you for submitting your manuscript to PLOS ONE. After careful consideration, we feel that it has merit but does not fully meet PLOS ONE’s publication criteria as it currently stands. Therefore, we invite you to submit a revised version of the manuscript that addresses the points raised during the review process.

**ACADEMIC EDITOR: One or more of the reviewers has recommended that you cite specific previously published works. Members of the editorial team have determined that the works referenced are not directly related to the submitted manuscript. As such, please note that it is not necessary or expected to cite the works requested by the reviewer**

We look forward to receiving your revised manuscript.

Kind regards,

Mahmood Ahmed

Academic Editor

PLOS ONE

Journal requirements: When submitting your revision, we need you to address these additional requirements. 1. Please ensure that your manuscript meets PLOS ONE's style requirements, including those for file naming. The PLOS ONE style templates can be found at https://journals.plos.org/plosone/s/file?id=wjVg/PLOSOne_formatting_sample_main_body.pdf and https://journals.plos.org/plosone/s/file?id=ba62/PLOSOne_formatting_sample_title_authors_affiliations.pdf 2. In your Methods section, please provide additional information regarding the permits you obtained for the work. Please ensure you have included the full name of the authority that approved the field site access and, if no permits were required, a brief statement explaining why. 3. We suggest you thoroughly copyedit your manuscript for language usage, spelling, and grammar. If you do not know anyone who can help you do this, you may wish to consider employing a professional scientific editing service.  The American Journal Experts (AJE) (https://www.aje.com/) is one such service that has extensive experience helping authors meet PLOS guidelines and can provide language editing, translation, manuscript formatting, and figure formatting to ensure your manuscript meets our submission guidelines. Please note that having the manuscript copyedited by AJE or any other editing services does not guarantee selection for peer review or acceptance for publication.  Upon resubmission, please provide the following: The name of the colleague or the details of the professional service that edited your manuscript A copy of your manuscript showing your changes by either highlighting them or using track changes (uploaded as a *supporting information* file) A clean copy of the edited manuscript (uploaded as the new *manuscript* file)”. 4. We note that your Data Availability Statement is currently as follows: [All relevant data are within the manuscript and its Supporting Information files.] Please confirm at this time whether or not your submission contains all raw data required to replicate the results of your study. Authors must share the “minimal data set” for their submission. PLOS defines the minimal data set to consist of the data required to replicate all study findings reported in the article, as well as related metadata and methods (https://journals.plos.org/plosone/s/data-availability#loc-minimal-data-set-definition). For example, authors should submit the following data: - The values behind the means, standard deviations and other measures reported;- The values used to build graphs;- The points extracted from images for analysis. Authors do not need to submit their entire data set if only a portion of the data was used in the reported study. If your submission does not contain these data, please either upload them as Supporting Information files or deposit them to a stable, public repository and provide us with the relevant URLs, DOIs, or accession numbers. For a list of recommended repositories, please see https://journals.plos.org/plosone/s/recommended-repositories. If there are ethical or legal restrictions on sharing a de-identified data set, please explain them in detail (e.g., data contain potentially sensitive information, data are owned by a third-party organization, etc.) and who has imposed them (e.g., an ethics committee). Please also provide contact information for a data access committee, ethics committee, or other institutional body to which data requests may be sent. If data are owned by a third party, please indicate how others may request data access. 5. Please include your tables as part of your main manuscript and remove the individual files. Please note that supplementary tables (should remain/ be uploaded) as separate ""supporting information"" files"".

Additional Editor Comments:

Major revision

Reviewers' comments:

Reviewer's Responses to Questions

**Comments to the Author**

1. Is the manuscript technically sound, and do the data support the conclusions?

Reviewer #1: No

Reviewer #2: Yes

Reviewer #3: Yes

2. Has the statistical analysis been performed appropriately and rigorously? 

Reviewer #1: No

Reviewer #2: No

Reviewer #3: Yes

3. Have the authors made all data underlying the findings in their manuscript fully available?

Reviewer #1: Yes

Reviewer #2: Yes

Reviewer #3: Yes

4. Is the manuscript presented in an intelligible fashion and written in standard English?

Reviewer #1: No

Reviewer #2: Yes

Reviewer #3: Yes

5. Review Comments to the Author

Reviewer #1: Redox cycling of sulfur via microbes in soil boosts the bioavailability of nutrients to Brassica napus

# Title: OK

Abstract: Rewriting with numerical details required.

KEYWORDS: Alphabetical rearrangement suggested.

Introduction:

Line 58-62: simple, lucid and complete sentences will be appreciated.

Statement of novelty needed.

2.1 Soil characterization

SRM data need to be disclosed.

Details of studied crop required.

2.8 Analysis of soil, 2.9 Plant analysis: SRM validation is a must. Do the needful.

Number of treatment combination, design, replication need to be disclosed in statistics.

Correlational aspect of Mn, Zn, Fe and S in terms of crop growth needs additional attention with current citations.

Authors are required to include (i) how the farmers will be benefited from the study and (ii) consequences of excess S application on PK cycle.

Table 1. Effect of microbial inoculants and fertilizer application on crop growth parameters.

Table 2-6. Effect of microbial inoculants and fertilizer application on plant macronutrients

Data should be accompanied by letter cases.

Some data should be in figure format

## English check suggested

Reviewer #2: the present study is very informative and of practical use. i congratulate the authors for working on this issue.

the paper need revision before acceptance for publication. the queries have been marked in the paper.

Reviewer #3: The manuscript presents a relevant study on the redox cycling of sulfur via microorganisms in soil, focusing on increasing nutrient bioavailability for canola cultivation. The research was conducted rigorously, with well-structured experiments, appropriate controls, and statistical analysis. The results obtained are solid and support the conclusions that the inoculation of microorganisms (SOB and SRB) along with chemical fertilizers improves plant growth and nutrient bioavailability.

However, some points could be improved:

Introduction Review:

Example of repetition: In the paragraph discussing the importance of sulfur for plant nutrition, there is a repetition of ideas about sulfur’s role in proteins and photosynthesis. The text could be more concise, addressing these points more directly. For example:

"Sulfur is essential for the synthesis of amino acids such as methionine and cysteine, which are crucial for plant growth, as well as playing important roles in the formation of proteins and vitamins." This could be condensed to: "Sulfur is vital for the formation of amino acids like methionine and cysteine, which are fundamental for plant growth and the synthesis of proteins and vitamins."

Discussion:

Example of critical analysis: The discussion mentions that the inoculation of SOB and SRB increased nutrient bioavailability, but it does not address comparisons with other studies that may show different results. It would be interesting to discuss, for instance:

"Although this study showed clear benefits of SOB and SRB inoculation, other studies, such as [Author et al., year], found limited results with similar microorganisms in acidic soils. The variation in results may be attributed to differences in soil characteristics and environmental conditions."

Grammatical and Linguistic Errors:

Example of grammatical error: In some parts, there are phrases that could be clearer. For example, in a section:

"The applicability of the results obtained are evident to increase agricultural productivity." The correct form would be: "The applicability of the results obtained is evident in increasing agricultural productivity." Additionally, phrases like "The effect of SOB and SRB on plants were significant" should be revised to "The effect of SOB and SRB on plants was significant," correcting the subject-verb agreement error.

Supporting Data:

Example of raw data: If the authors mention that SOB and SRB inoculation increased nutrient concentrations like nitrogen (N) and phosphorus (P), it would be helpful to include the raw numeric values, such as:

"The nitrogen concentration in the plant tissue was 1.53% in the ½NPK+SOB+SRB combination." These data should be presented in more detail, perhaps in a supplemental table or a publicly accessible data repository.

Conclusion:

Example of a more robust conclusion: The conclusion could be more detailed by addressing the study’s limitations and suggesting future research areas. An example would be:

"While the results clearly showed the benefits of SOB and SRB inoculation, it would be interesting to conduct field studies to evaluate the effectiveness of these inoculations in different soil types and under varying climatic conditions. Furthermore, the interaction between different microorganisms and fertilizers could be explored in future research to optimize resource use in canola cultivation."

6. PLOS authors have the option to publish the peer review history of their article (what does this mean? ). If published, this will include your full peer review and any attached files.

**Do you want your identity to be public for this peer review?** For information about this choice, including consent withdrawal, please see our Privacy Policy .

Reviewer #1: No

Reviewer #2: **Yes: ** Vivek Sharma

Reviewer #3: No

---

## [Author Response · Author response to Decision Letter 1]

12 Dec 2024

Response to Reviewers’ Comments

Reference: PONE-D-24-23262

Title: Redox cycling of sulfur via microbes in soil boosts the bioavailability of nutrients to Brassica napus

NOTE: Revised/added text as per review comments has been highlighted in the revised manuscript through track changes.

Authors acknowledge the comments by respectable editor and reviewers to improve the quality of the manuscript. Thank you very much for these efforts and for sparing valuable time.

Reviewer #1:

Query 1. In the abstract numerical details are required.

Response: Numerical data has been added in the abstract as per suggestion.

Query 2. Keywords: Alphabetical rearrangement is suggested.

Response: Keywords have been arranged in alphabetical order as suggested.

Query 3. Line 58-62: simple, lucid, and complete sentences will be appreciated. Statement of novelty needed.

Response: Needful has been done and a statement of novelty is added in lines.

Query 4. Provide analysis of soil/characterization and plant analysis. The number of treatment combinations, design, and replications need to be disclosed in statistics.

Response: The initial data related to soil have been provided under subheading 2.1 Soil characterization. Plant analysis has been described under subheading 2.9 Plant analysis. The number of treatment combinations, design, and number of replications has been added as suggested under subheading 2.10 Statistical analysis however, the same information is available in subheading, 2.6 Treatment plan, and experimental design

Query 5. Authors are required to include (i) how the farmers will benefit from the study and (ii) the consequences of excess S application on the PK cycle.

Response: Suggestions have been incorporated in the last paragraph of the discussion.

Query 6. Table 1. Effect of microbial inoculants and fertilizer application on crop growth parameters. Table 2-6. Effect of microbial inoculants and fertilizer application on plant macronutrients. Data should be accompanied by letter cases. Some data should be in figure format.

Response: All tables include alphabets showing significant differences. Additionally, the plant macronutrient data (Table 2) has been converted into a graph (Fig 1), as suggested.

Query 7. English check suggested.

Response: Needful has been done.

Reviewer #2:

Query 1. Line 23-23: In the first phase isolation of S bacteria was carried out while in the second phase, results are missing.

Response: Data related S bacteria has been provided under subheading 3.1 SOB and SRB isolate attributes. In this step, only characterization of isolates was carried out.

Query 2. KSO4-2

Response: Line 173 corrected the typing error K2SO4.

Query 3. The conclusion should be concrete. (% increase in yield and nutrient availability with best treatment).

Response: Suggestion incorporated.

Query 4. Isolation and Characterization of SOB and SRB, results not given.

Response: Data has been provided under subheading 3.1 SOB and SRB isolate attributes.

Query 5. All the references have been updated and formatted as the journal format.

Response: Needful has been done.

Query 6. Alphabets showing significant differences are missing in the table

Response: Alphabets showing significant differences are included in all tables and Figure as per suggestion.

Reviewer #3:

Query 1. Example of repetition: In the paragraph discussing the importance of sulfur for plant nutrition, there is a repetition of ideas about sulfur’s role in proteins and photosynthesis. The text could be more concise, addressing these points more directly.

Response: Suggestion incorporated.

Query 2. Example of critical analysis: The discussion mentions that the inoculation of SOB and SRB increased nutrient bioavailability, but it does not address comparisons with other studies that may show different results.

Response: Although the related data is very limited, however, a few relevant data are included in the discussion section.

Query 3. Example of grammatical error: In some parts, some phrases could be clearer. For example, in a section:

"The applicability of the results obtained are evident to increase agricultural productivity." The correct form would be: "The applicability of the results obtained is evident in increasing agricultural productivity." Additionally, phrases like "The effect of SOB and SRB on plants were significant" should be revised to "The effect of SOB and SRB on plants was significant," correcting the subject-verb agreement error.

Response: Suggestion incorporated.

Query 4. Supporting Data:

Example of raw data: If the authors mention that SOB and SRB inoculation increased nutrient concentrations like nitrogen (N) and phosphorus (P), it would be helpful to include the raw numeric values, such as: "The nitrogen concentration in the plant tissue was 1.53% in the ½NPK+SOB+SRB combination." These data should be presented in more detail, perhaps in a supplemental table or a publicly accessible data repository.

Response: Needful has been done.

Query 5. The conclusion could be more detailed.

Response: The conclusion has been updated as per suggestion.

---

## [Decision Letter · Decision Letter 1]

24 Jan 2025

Redox cycling of sulfur via microbes in soil boosts the bioavailability of nutrients to Brassica napus

PONE-D-24-23262R1

Dear Dr. Alam,

We’re pleased to inform you that your manuscript has been judged scientifically suitable for publication and will be formally accepted for publication once it meets all outstanding technical requirements.

Kind regards,

Mahmood Ahmed

Academic Editor

PLOS ONE

Additional Editor Comments (optional):

Reviewers' comments:

Reviewer's Responses to Questions

**Comments to the Author**

1. If the authors have adequately addressed your comments raised in a previous round of review and you feel that this manuscript is now acceptable for publication, you may indicate that here to bypass the “Comments to the Author” section, enter your conflict of interest statement in the “Confidential to Editor” section, and submit your "Accept" recommendation.

Reviewer #3: All comments have been addressed

2. Is the manuscript technically sound, and do the data support the conclusions?

Reviewer #3: Yes

3. Has the statistical analysis been performed appropriately and rigorously? 

Reviewer #3: Yes

4. Have the authors made all data underlying the findings in their manuscript fully available?

Reviewer #3: Yes

5. Is the manuscript presented in an intelligible fashion and written in standard English?

Reviewer #3: Yes

6. Review Comments to the Author

Reviewer #3: (No Response)

7. PLOS authors have the option to publish the peer review history of their article (what does this mean? ). If published, this will include your full peer review and any attached files.

**Do you want your identity to be public for this peer review?** For information about this choice, including consent withdrawal, please see our Privacy Policy .

Reviewer #3: No

---

## [Editor Report · Acceptance letter]

PONE-D-24-23262R1

PLOS ONE

Dear Dr. Alam,

I'm pleased to inform you that your manuscript has been deemed suitable for publication in PLOS ONE. Congratulations! Your manuscript is now being handed over to our production team.

Kind regards,

on behalf of

Dr. Mahmood Ahmed

Academic Editor

PLOS ONE